

# Roles of Tropical Waves in the Formation of Global Tropical Cyclone Clusters

The-Anh Vu[1] and Chanh Kieu[1,*]

[1]Department of Earth and Atmospheric Sciences, Indiana University, Bloomington, IN, 47405

**Correspondence:** Department of Earth and Atmospheric Sciences, Indiana University, Bloomington, IN, 47405. Email: ckieu@indiana.edu

**Abstract.** This study examines the role of tropical dynamics in the formation of global tropical cyclone (TC) clusters. Using theoretical analyses and idealized simulations, it is found that global TC clusters can be produced by the internal dynamics of the tropical atmosphere, even in the absence of landmass surface and zonal sea surface temperature (SST) anomalies. Our analyses of a two-dimensional InterTropical Convergence Zone (ITCZ) model capture indeed some planetary-scale stationary

modes whose zonal and meridional structures can support the formation of TC clusters at the global scale. Additional idealized simulations using the Weather Research and Forecasting (WRF) model confirm these results in a range of aqua-planet experiments. Specifically, the examination of two common tropical waves including the equatorial Rossby (ER) wave and the equatorial Kelvin (EK) wave shows that ER waves could develop and maintain a planetary-scale stationary structure for a range of zonal wavenumbers $[5-11]$, while EK waves do not. This numerical result is consistent with the ITCZ breakdown model

and reveals some forcing structures that can support stationary "hot spots" for global TC formation. The findings in this study offer different insights into the importance of tropical waves in producing global TC clusters beyond the traditional explanation based on zonal SST anomalies.

## 1   Introduction

It has been well observed that global tropical cyclone (TC) formation prefers some specific regions where more TCs form than other areas. Within the tropical region ($\sim$[25S-25N]), several main TC clusters can be identified, which include the northeastern and northwestern Pacific, the north Atlantic, the north and south Indian, and the southwestern Pacific basins (Ramsay, 2017; Strachan et al., 2013; Knapp et al., 2010). From the global perspective, the emergence of such TC clusters is ultimately linked to the current Earth's terrain and landmass arrangement, which accords with previous modeling studies of global TC climatology

in the presence of a realistic landmass pattern (e.g., Bell et al., 2013; Wehner et al., 2017; Yoshimura et al., 2006; Kieu et al., 2023). Indeed, various aqua-planet simulations have shown that the clustering of global tropical cyclogenesis (TCG) does not





exist in the absence of landmass (e.g., Ballinger et al., 2015; Viale and Merlis, 2017; Chavas and Reed, 2019; Vu et al., 2021; Zhang et al., 2021), thus confirming the key role of the land surface in the current TC climatology.

From a broad perspective, the most direct effect of the land-sea arrangement on global TC climatology is via the distribution of sea surface temperature (SST), whose zonal anomalies are considered to be the leading factor for TC clusters (Zhou et al., 2014; Viale and Merlis, 2017; Fedorov et al., 2019; Walsh et al., 2020). Specifically, SST zonal anomalies result in different surface fluxes in different ocean areas and directly impact the air-sea interaction that governs TC development and related climatology (Emanuel, 2018; Knutson et al., 2008, 2010). On the other hand, irregular SST distribution can indirectly modulate large-scale tropical circulations that affect TCG via environmental controls such as vertical wind shear, low-level moist convergence, or the tropospheric static stability (Zhang et al., 2021; Kossin and Vimont, 2007; Ferrara et al., 2017; Downs and Kieu, 2020; Kieu et al., 2023), which are often represented by some climate indices such as the potential genesis index (GPI) (e.g., Emanuel and Nolan, 2004; Camargo and Wing, 2016; Menkes, 2012). The combined effects of zonally anomalous surface fluxes and tropical circulations dictate the overall TCG climatology as currently observed.

From the dynamical standpoint, global TC clusters could also be manifestations of tropical dynamics, which may contain planetary-scale wave modes that set up specific favorable "hot spots" for TCG, even in the absence of all zonal SST variations. Numerous observational and modeling studies have shown that large-scale tropical waves can provide an efficient pathway for TC formation as discussed, e.g., in Molinari et al. (1997); Kieu and Zhang (2009, 2010); Dunkerton et al. (2009); Wang et al. (2019, 2014). Among various types of tropical wave-like disturbances, several have been identified to be particularly prone to TCG. These wave-like disturbances exist in both the tropical atmosphere and ocean and include, e.g., tropical Kelvin, equatorial Rossby (ER), mixed Rossby-gravity, and westward and eastward inertio-gravity waves. In general, their typical scale varies between the synoptic and planetary ranges (5000-10000 km), with a frequency of $\sim$ 5-10 days and phase speed from 10 to 25 $ms^{-1}$ (Takayabu and Nitta, 1993; Molinari et al., 1997; Dunkerton and Baldwin, 1995; Holton, 2004).

Using outgoing longwave radiation data and global reanalysis winds, Frank and Roundy (2006) could further classify different tropical waves, based on their impacts on TCG. Their results showed that the enhanced activities of ER, mixed Rossby-gravity waves, tropical depressions, and Madden-Julian Oscillation (MJO) are strongly linked to TCG in most ocean basins. These wave activities can augment environmental vorticity and modulate vertical wind shear in TCG-favorable areas. Similarly, Bessafi and Wheeler (2006) examined the roles of different tropical waves in relation to TC activities over the South Indian Ocean, and concluded that MJO and ER waves tend to modulate TCG, whereas Kelvin and mixed Rossby-gravity waves play a less significant role. Using rainfall anomaly, Schreck et al. (2011) demonstrated that TCG could be linked to equatorial waves after properly filtering unrelated anomalies.



While tropical waves can play an essential role in modulating TCG as documented in previous studies, our understanding of their roles relative to other mechanisms is still limited. Using observational analyses, Dunkerton et al. (2009) discussed four different pathways for TCG including (i) tropical waves, (ii) intertropical convergence zone (ITCZ) or monsoon troughs, (iii) extratropical disturbances and (iv) topographic flows. These different pathways are generally hard to isolate in real atmospheric

conditions due to their mutual roles in TC formation. For instance, African waves have been shown to account for almost half of hurricanes in the North Atlantic (Avila and Pasch, 1992; Frank and Clark, 1980, e.g,), while more substantial impacts of MJO on TCG are observed over the North Pacific ocean (Molinari et al., 2000; Maloney and Hartmann, 2000) and Australia region (Hall et al., 2001). In the northwestern Pacific basin, large-amplitude tropical wave packets related to off-equatorial tropical depression TD-type disturbances are however more abundant as reported in, e.g., Dickinson and Molinari (2002). Thus, the

relative importance of different pathways are generally ad-hoc and basin-specific.

Among all TCG pathways, we note that ITCZ and tropical waves are intrinsic to tropical dynamics, as they can exist even in the absence of all landmass or terrain effects. Indeed, extensive idealized modeling studies have shown that TCG occurs naturally in the aqua-planet setting without any terrain-induced disturbances or monsoon troughs (Reed and Jablonowski, 2011; Merlis et al., 2013; Vu et al., 2021). So long as a model can capture the characteristics of the ITCZ, TC climatology can

display an expected behavior at the global scale. In this regard, the aqua-planet simulations suggest that TCG processes should be an inherent part of large-scale tropical dynamics.

Given the aforementioned role of tropical waves and the ITCZ in global TCG, how their interaction could produce a favorable environment for the formation of TCG clusters is still open. For example, an early study by Charney and DeVore (1979) showed that planetary-scale baroclinic modes may experience blocking effects under some specific terrain forcings. This blocking

creates a recurrence pattern that has many implications for large-scale flows such as the formation of midlatitude cyclones (e.g., Charney and DeVore, 1979; Legras and Ghil, 1985; Rambaldi and Mo, 1984) or stationary waves in the Northern Hemisphere (e.g., Ting, 1994; Wills et al., 2019; Wang et al., 2020). Such stationary patterns with a horizontal scale of $\sim \mathcal{O}(10^5)$ km can in fact interact with tropical easterlies and affect global TCG as recently proposed in Wang et al. (2020).

A natural question that we wish to present in this study is how planetary-scale stationary waves (PSW) could set up favorable

conditions for global TC clusters beyond the direct impacts of SST zonal anomalies. Note that planetary-scale stationary modes in the tropics are generally different from tropical waves that are often considered to be a source of tropical disturbances (e.g., Frank, 1977; TaiOgura, 1987; Zehnder et al., 1999). While tropical waves propagate in time, PSWs only vary spatially and so they can provide favorable environments for TCG in some specific areas. In this study, we will examine whether these PSWs can actually exist in the tropics and how they affect the formation of global TCG clusters. By carrying out theoretical analyses



along with numerical simulations, it is expected that the internal dynamics of these PSWs and their role in the clustering of global TCG can be isolated and better understood.

The rest of this study is organized as follows. Section 2 presents an analytical model for PSWs in the tropics that could support the clustering of global TC formation. Section 3 provides detailed descriptions of our numerical designs and related results to verify the theoretical analyses in Section 2. A summary of key findings is then provided in the final section.

## 2   Theoretical analyses

To examine the existence of PSWs in the tropics, we present first in this section a theoretical framework based on the ITCZ breakdown model. Our choice of the ITCZ model herein stems from the observation that the ITCZ is a dominant mechanism of tropical dynamics in the absence of landmass or zonal SST anomalies (Zehnder et al., 1999; Molinari et al., 2000; Ferreira and Schubert, 1997; Wang and Magnusdottir, 2006; Kieu and Zhang, 2010; Wang et al., 2019; Vu et al., 2021). The barotropic

instability zone along the ITCZ where horizontal winds from the two hemispheres converge creates a unique band with strong horizontal shear that ensures the Rayleigh instability. As such, perturbations embedded within the ITCZ would likely grow by extracting the energy from the background, leading to the development of TC clusters associated with the ITCZ (Charney and Stern, 1962; Ferreira and Schubert, 1997; Holton, 2004).

Following Wang et al. (2019, hereinafter W19), we employ here an ITCZ breakdown model that is based on the shallow

water model proposed by Ferreira and Schubert (1997). The constraint on the horizontal flows of this ITCZ model allows one to obtain a single governing equation for the horizontal streamfunction $\psi$ as follows (see W19)

$$\frac{d\Delta\psi}{dt} = \nu_e\Delta^2\psi + F - \alpha\Delta\psi - \beta\frac{\partial\psi}{\partial x}, \tag{1}$$

where $F$ is an external force that represents the mass source/sink within the ITCZ, and $\nu_e$ and $\alpha$ are the horizontal eddy viscosity coefficient and relaxation time, [1], respectively. Note that the external forcing term $F$ is important for the ITCZ model,

because the horizontal dynamics could not capture the vertical mass flux inside the ITCZ. As such, the inclusion of the forcing $F$ is required to properly represent the ITCZ dynamics.

For our global TCG problem, a zonal tropical domain $\Omega = [0, L_x] \times [0, L_y]$ is applied, where $L_y$ is the width of the tropical channel and $L_x$ is the zonal length of the channel similar to that used in W19's study. This domain roughly represents a region

---

[1]In Charney and DeVore (1979), the relaxation time $\alpha$ is proportional to the ratio of the Ekman depth $D_E$ over the depth of the fluid $H$, while the forcing term $F$ is treated as an external large-scale vorticity source.





**Table 1.** The parameters and their corresponding scales for the ITCZ breakdown model

| Parameters | Remark |
| --- | --- |
| $\beta \sim 10^{-11}\ m^{-1}s^{-1}$ | beta parameter in the tropical region |
| $\nu_e \sim 10 - 10^4\ m^2 s^{-1}$ | the horizontal eddy viscosity |
| $\alpha \sim 10^{-5} - 10^{-7}\ s^{-1}$ | the large-scale relaxation time inverse |
| $\gamma \sim 10^{-10} - 10^{-11}\ s^{-2}$ | Kolmogorov forcing amplitude for mass sink/source |
| $L_x \sim 4 \times 10^4\ km$ | the zonal scale of the tropical channel |
| $L_y \sim 1.5 \times 10^3\ km$ | the meridional scale of the tropical channel |
| $U_0 \sim 10 - 20\ ms^{-1}$ | the zonal scale of the low-level zonal wind component in the tropics |
| $\epsilon \sim 0.4 - 0.8$ | the Rossy number |
| $E \sim 3 \times 10^{-7} - 10^{-3}$ | the Ekman number |
| $A \sim 4 \times 10^{-3} - 10^{-1}$ | the nondimensional ratio of large-scale relaxation and Coriolis parameter |

where the ITCZ can be treated as a long band wrapping around the equator. For the current Earth's condition, $L_x \sim 40,000$

km, and $L_y \sim 1,200 - 1,500$ km (i.e., a width of 12-15$^o$), and so by construction $L_x \gg L_y$.

Because we focus on the ITCZ dynamics, the forcing term $F$ must be given *a priori*. Here, we will follow W19 and Pan et al. (2021) and employ the Kolmogorov forcing such that its steady-state can best capture the typical background flows associated with the ITCZ in the tropical lower troposphere as follows:

$$F = \gamma \sin \frac{\pi y}{L_y} \tag{2}$$

where $\gamma$ denotes the amplitude of the mass forcing. Note that $\gamma$ cannot be arbitrary, because its value dictates the magnitude of the ITCZ's zonal mean flow in the tropical region. For a typical zonal flow in the tropic with $U_0 = \mathcal{O}(10)\ ms^{-1}$, $\gamma$ is therefore $\approx 10^{-11}\ s^{-2}$ (see Table 1 for descriptions and magnitudes of all parameters in the ITCZ model).

Among different forcing functions, the Kolmogorov forcing is of special interest for our investigation here because it produces a steady solution $\psi_S$ consistent with the typical flow near the ITCZ, which is given by

$$\psi_S = \frac{-\gamma L_y^4}{\nu_e \pi^4 + \alpha L_y^2 \pi^2} \sin \frac{\pi y}{L_y}. \tag{3}$$

The corresponding meridional profile of the zonal flow $U_s(y)$ obtained from this steady state is therefore

$$U_s(y) = U_0 \cos \frac{\pi y}{L_y}, \text{ where } U_0 \equiv \frac{\pi \gamma L_y^3}{\nu_e \pi^4 + \alpha L_y^2 \pi^2} \tag{4}$$



This zonal steady state possesses an easterly flow and a westerly flow to the north and the south of the ITCZ that is located at $y = L_y/2$ as expected during the Northern Hemisphere summer (see Figure 1 in W19).

With the Kolmogorov forcing $F$ and its corresponding steady state, our problem now is to search for the conditions on the model parameters that permit PSWs. We note that our focus on the stationary structure in this study, rather than the dynamical transition of the ITCZ model as in W19, is because these structures could produce large-scale favorable regions for TC formation. If these regions are stationary, they can account for the formation of global TC clusters that we wish to examine herein.

For the sake of convenience, we first re-write Eq. (1) in a nondimensional form using the following set of scales for time, distance, and streamfunction:

$$t = \frac{1}{L_y\beta}t^*, \ \ \psi = LU_0\psi^*, \ \ (x,y) = L_y(x^*,y^*), \ \ F = \frac{\alpha U_0}{L_y}F^*,$$

where the asterisk denotes the nondimensional variables. Linearizing the streamfunction around the steady state as $\psi^* = \psi_S^* + \psi'$ leads to

$$\frac{\partial \Delta \psi'}{\partial t} = E\Delta^2\psi' - I\Delta\psi' - \frac{\partial\psi'}{\partial x} + R\frac{d\widetilde{\psi_S}}{dy}\frac{\partial\Delta\psi'}{\partial x} - R\frac{d^3\widetilde{\psi_S}}{dy^3}\frac{\partial\psi'}{\partial x}, \tag{5}$$

where

· $E = \frac{\nu_e}{L^3\beta}$ is the Ekman number;

· $I = \frac{\alpha}{L\beta}$ is the nondimensional ratio of the eddy relaxation time to the time scale related to the Earth's rotation;

· $R = \frac{\gamma_1\epsilon}{E\pi^3 + I\pi}$ is the Rayleigh number;

· $\widetilde{\psi_S} = -\frac{\sin\pi y}{\pi}$ is the meridional dependence of the steady state; and

· $\epsilon = \frac{U_0}{L^2\beta}$ is the Rossby number;

Similar to W19, we also employ the same mathematical procedure to simplify our analyses by extending the tropical domain from $[0, L_y]$ to $[-L_y, L_y]$. This domain extension has the benefit of symmetrizing the meridional boundary conditions as discussed in W19. In the nondimensional form, this extended domain becomes

$$\Omega = \left[0, \frac{2}{a}\right] \times [-1, 1]. \tag{6}$$





where the scale factor $a \equiv 2L_y/L_x$. Because of the nature of tropical waves, we apply a periodic boundary condition to the zonal direction but a free boundary condition in the meridional direction for Eq. (5) on the domain $\Omega$, i.e.,

$$\psi'(t,0,y) = \psi'\left(t, \frac{2}{a}, y\right), \ \psi'(t,x,-1) = \psi'(t,x,1) = \frac{\partial^2 \psi'}{\partial y^2}(t,x,-1) = \frac{\partial^2 \psi'}{\partial y^2}(t,x,1) = 0. \tag{7}$$

Note that the free boundary condition at $y = -1$ and $y = 1$ will ensure that no exchange along the north/south meridional

boundaries will be allowed (i.e., no $v$-wind component at $y = -1$ and $y = 1$).

Given the governing Eq. (5) for $\psi'$ and boundary conditions (7), we can now examine the existence of stationary states that help set up favorable hot spots for global TC formation. Here, a stationary state of any perturbation is defined as the one that is independent of time but still varies in spatial directions (see, e.g., Holton, 2004, Chapter 7). Specifically, we will look for solutions of the form

$$\psi'_m(x,y) = e^{i\pi m a x} \Psi(y), \tag{8}$$

where $m \in \mathbb{Z}$ represents zonal wavenumbers and the zonally–periodic boundary condition has been implicitly used. Substitute this solution into Eq. (5), it can be seen that $\Psi(y)$ must satisfy

$$E(D^2 - a^2 m^2 \pi^2)^2 \Psi - I(D^2 - a^2 m^2 \pi^2)\Psi - ima\pi\Psi + ima\pi RD(\widetilde{\psi}_S)(D^2 - a^2 m^2 \pi^2)\Psi - ima\pi RD^3(\widetilde{\psi}_S)\Psi = 0 \tag{9}$$

with boundary conditions

$$\Psi(-1) = \Psi(1) = \frac{d^2 \Psi(-1)}{dy^2} = \frac{d^2 \Psi(1)}{dy^2} = 0, \tag{10}$$

Because of (10), $\Psi(y)$ can be further expressed in the following form

$$\Psi(y) = \sum_{n \geq 0} \phi_n \cos\left(n + \frac{1}{2}\right)\pi y + \sum_{n \geq 1} \widetilde{\phi}_n \sin n\pi y, \tag{11}$$



where $\phi_n$ and $\widetilde{\phi}_n$ are the coefficients to be determined by Eq. (9). Putting together Eqs. (8) and (9), the stationary solution $\psi_m(x,y)$ is then given by

$$\psi'_m(x,y) = \sum_{n\geq 0} i^n e^{ima\pi x}\phi_{m,n}\cos\left(n+\frac{1}{2}\right)\pi y + \sum_{n\geq 1} i^n e^{ima\pi x}\widetilde{\phi}_{m,n}\sin n\pi y, \quad m\in\mathbb{Z}. \tag{12}$$

As shown in W19, imposing the physical requirement on the existence of the wave-like solutions (i.e., $\phi_{m,n}\neq 0$ and $\widetilde{\phi}_{m,n}\neq 0$) can provide great insights into the stability and structure of the wave solutions. Recall that we examine herein the existence of PSW with nonzero amplitudes, and so it can be seen readily from Eqs. (9)-(11) that there exist no stationary waves for zonal wavenumber $m=0$. For $m\neq 0$, it can be also derived directly from Eqs. (9)-(12) that all possible stationary waves with $m\neq 0$
must satisfy the following conditions (Appendix A)

$$\begin{cases} \phi_{m,n}=0, & \text{when } \frac{\sqrt{3}}{2K}\leq a < \frac{\sqrt{3}}{2K-2}, \ |m|>K, \ K=1,2,3...,\forall n \\ \widetilde{\phi}_{m,n}=0, & n\geq 1. \end{cases} \tag{13}$$

where $K$ is the lower bound for $m$ above which all stationary modes cannot exist (i.e., $K$ is an upper bound for the zonal wavenumbers below which stationary wave modes can exist). Given the aspect ratio $a\sim 0.054$ in the Earth's current tropical atmosphere, $K$ is $\sim 11$. As such, any possible PSW in the tropics must have a zonal wavenumber of less than 11.
This upper bound limit $K$ for PSWs will have a profound implication on the clustering of global TCG as shown later. To see the significance of this result for now, consider a hypothetical atmosphere for which the external forcing could support an upper bound $K=2$. The only stationary mode allowed by this upper bound is $m=1$, whose structure can be shown to be of the form (see Appendix A)

$$\psi'(x,y) = \left(\phi_1\cos(\frac{\pi y}{2}) + \phi_2\cos(\frac{3\pi y}{2})\right)\sin(2\pi x), \tag{14}$$

where $\phi_{1,2}$ are the wave amplitudes. This solution demonstrates that any PSW in the tropical region cannot possess a structure that is monotonically decreasing with latitudes. This eliminates certain types of tropical waves for PSWs such as equatorial Kelvin waves. In this regard, (14) imposes a strong constraint on both the meridional and the zonal structures of any PSW in the tropics.

The second significant implication from the above result is that not only do PSWs exist, but they are guaranteed to be
stable when the Rayleigh number $R$ is sufficiently smaller than a critical limit $R_m^*$. As presented in Pan et al. (2021), the



critical Rayleigh number $R_m^*$ is generally a function of the zonal wavenumber $m$ as well as other characteristics of the tropical atmosphere such as the Ekman number, the Rossby number, or mean flow $U_0$, which dictate the Hopf bifurcation of the ITCZ model. While the constraint (13) could give us information about the upper bound on the zonal wavenumber for the existence of PSWs, it is not generally known in advance what zonal wavenumber $m$ will emerge. That is, all wavenumbers $m = 1...K$ are possible stationary modes in the tropical region. In this regard, the constraint (13) is merely a necessary condition for the existence of PSWs. The exact stationary wavenumber that can emerge must be further determined by the Rayleigh number $R$ to ensure the structural stability of PSWs (see Figure 2 in W19 and discussion therein for the characteristics of $R_m^*$), which is however beyond the scope of this work.

Despite the lack of stability analyses for the stationary modes obtained in the above analyses, our results on the upper bound for the stationary zonal wavenumbers and their possible structure still suggest very specific types of stationary structure in the tropics. In the next section, we will verify these PSWs and their implication for the clustering of global TCG, using numerical simulations.

## 3 Numerical experiments

### 3.1 Tropical wave initialization

As shown in the previous section, planetary-scale stationary modes in the tropics must possess some particular meridional and zonal structures for their existence. This important result motivates us to implement different types of tropical waves in a numerical model to see whether PSWs can develop and how they support the formation of TCG clusters. Specifically in this section, we will verify if PSWs can exist under the same idealized aqua-planet setting as in the ITCZ model in Section 2, using numerical simulations.

In the first set of experiments, equatorial Kelvin (EK) waves are used to initialize the Weather Research and Forecasting (WRF) model (Skamarock and Coauthors, 2008, Version 3.9), similar to what was designed in Vu et al. (2021). For this purpose, the EK wave solution derived from the shallow water model is chosen herein, whose geopotential and horizontal flow perturbations are given by (see, e.g., Holton, 2004):

$$u(x,y) = U_0 e^{-\frac{\beta y^2}{2c_w}} e^{i(\omega t - kx)} + c.c. \tag{15}$$

$$v(x,y) = 0 \tag{16}$$

$$\phi(x,y) = c_w U_0 e^{-\frac{\beta y^2}{2c_w}} e^{i(\omega t - kx)} + c.c. \tag{17}$$



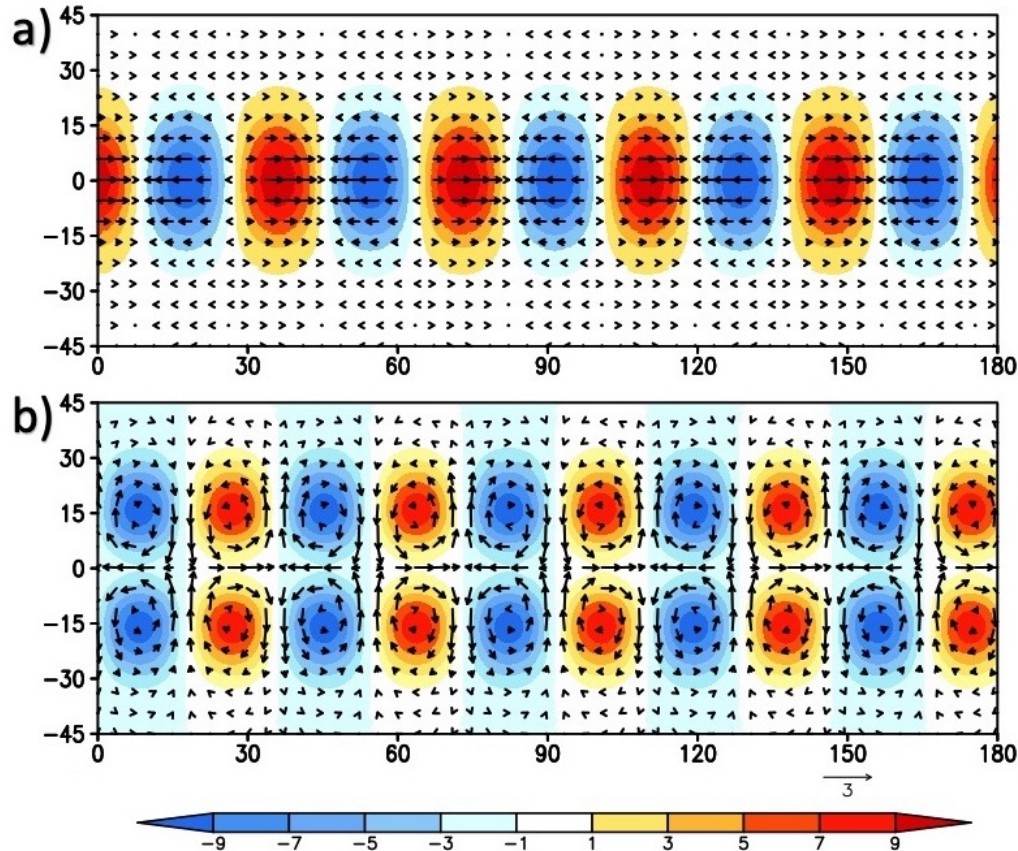

**Figure 1.** Illustration of the geopotential perturbation (shaded, unit: gpm) for (a) the EK wave and (b) the ER wave structure at 850 hPa with the zonal wavenumber $m = 10$, which is used to initialize the WRF model. Superimposed is the corresponding flow field (vectors) at the same level.

where $U_0$ is an initial wave amplitude, $c_w \equiv \omega/k$ is the EK phase speed, $\beta$ is the variation of the Coriolis parameter with latitude, and $y$ is the distance from the equator (Fig. 1a). Given an equivalent depth $H_e$, one can define the length scale $L$, the time scale $T$, and the EK wave speed $c_e$ as follows:

$$c_e = \sqrt{gH_e}, L = \sqrt{\frac{c_e}{\beta}}, T = (c\beta)^{-1/2} \tag{18}$$

210

Note that EK waves are chosen for our numerical experiments in this section because they possess a symmetric structure around the equator that monotonically decreases with latitudes. Because of this meridional profile, EK waves present a structure that is not sustainable for stationary modes as presented in Section 2. This way, one can verify if the theoretical results obtained from the ITCZ model about the non-existence of PSWs hold true for EK waves.





In the second set of experiments, equatorial Rossby (ER) waves were used, with the same model settings as for the EK wave experiment. These ER waves have a horizontal structure given also by the shallow water theory as follows (Kiladis et al., 2009).

$$v(x,y) = U_0 H(n,y) e^{-\frac{y^2}{2}} \cos(kx) \tag{19}$$

$$\phi(x,y) = -U_0 \frac{e^{-\frac{y^2}{2}}}{(\omega^2 - k^2)}[ky(n,y) - 2n\omega H(n-1,y) + \omega y(n,y)]\sin(kx) \tag{20}$$

$$u(x,y) = U_0 \omega^{-1}[k\phi(x,y) - yv(x,y)\tan(kx)] \tag{21}$$

where $H(n,y)$ is the Hermite polynomial, and $c_e$ is the ER phase speed. Unlike EK waves, ER waves possess a mass field with a meridional profile that peaks away from the equator as shown in Fig. 1b. Hence, they represent a type of structure that could support the development of stationary modes as presented in Section 2.

Beyond the motivation of choosing the above wave structures that could allow possible stationary states, our choice of ER and EK waves in this study is also motivated by previous results on the effects of these two wave types on TCG (Yamazaki and Murakami, 1989; Bessafi and Wheeler, 2006; Frank and Roundy, 2006; Molinari et al., 2007). For example, Yamazaki and Murakami (1989) found that TC formation is strongly linked to several zonal modes including the westward-propagating modes that possess similar characteristics with $n = 1$ ER waves. Likewise, Schreck (2015) found that the interaction of easterly waves and EK waves cans enhance the easterly wave convection and vorticity, resulting in more favorable conditions for TCG. From this perspective, these studies suggested that global TCG is connected to tropical waves, thus further motivating us to use EK and ER waves in our experiments.

In both EK and ER experiments, the wave amplitude $U_0$ is chosen to be 10 $ms^{-1}$, with a zonal wavenumber $m$ varying between $[5 - 15]$ (corresponding to a wavelength of $\sim$ 8000-3500km in the equatorial region) and a vertical structure given in terms of a normal mode (Wheeler, 2002; Wheeler et al., 2000). Note that the vertical normal mode is applied to both the wind and the mass fields in the ER and EK wave experiments. To ensure the balance requirement for other fields, we derive the temperature anomaly from geopotential anomaly, while the background moisture and temperature profiles are taken from Jordan's mean tropical sounding Jordan (1958).

Although there exists no strict balance dynamics in the equatorial region, the wave structure obtained from the shallow water dynamics could at least represent a consistent large-scale flow as evidenced by the minimal generation of smaller-scale gravity waves during the earlier period of simulations as reported in Gall et al. (2010). Therefore, it is anticipated that initial adjustment towards the model balance state can be quickly established during the first year of model integration. Of course, one can design in principle any tropical disturbance with an arbitrarily prescribed structure in numerical models. These waves are however



still expected to contain some balance between the mass and wind fields to minimize the model initial adjustment. Beyond the well-defined tropical wave theory derived from the shallow water system, there appears to have no existing theory that could provide a complete structure for tropical waves. As such, we will limit all of our experiments herein to the ER and EK wave

modes such that the WRF model's initial state can experience minimum model balance adjustment in our simulations.

### 3.2 Model configuration

Given the tropical wave initialization outlined in the previous section, a set of simulations for an aqua-planet tropical channel between [45S-45N] were conducted in this study. To cover the whole tropical region, a single domain with a homogeneous resolution of 27 km was used, with 1479×400 grid points in the west-east and the north-south directions and 31 vertical

levels. This domain is sufficiently large to capture important large-scale processes in the tropical as shown in V20, including the migration of TC tracks to higher latitudes and subsequent dissipation after transitioning to extratropical systems. While a higher horizontal resolution is required to fully capture the detailed inner-core dynamics of TCs, the 27-km resolution suffices to capture global TC formation and allow TCs to reach hurricane strength (Category 1 and above) as shown in previous studies (e.g., Zhao et al., 2009; Wehner et al., 2010; Kieu et al., 2023, V20). Because our focus is on global tropical cyclogenesis rather

than TC inner-core structure, this 27-km resolution was therefore employed herein.

For physical parameterization schemes, a set of model options similar to those used in V20 was chosen (Table 2), which capture well both the seasonality of the ITCZ and TC climatology. Note that the 1D ocean coupling option is needed to allow for proper air-sea feedback and seasonal variability, with a constant depth layer of 1 $m$ and a relaxation time of 30 days. Sensitivity experiments presented in V20 showed that ocean coupling is important for the ITCZ characteristics, as it dictates

the global TC climatology in the aqua-planet setting. For the purpose of this work, we used the same set of values for ocean parameters including the relaxation time and the mixed layer depth listed in Table 2, which best simulate the ITCZ dynamics as reported in V20.

### 3.3 Experiment design

All tropical wave simulations in this study were run for two years, starting from a mean state with either ER or EK wave

perturbations added to the model states as described in the previous section. Because the WRF model could spin up quickly and reach the model equilibrium within the first 3-6 months into integration, the first year of the simulation is treated as a spin-up period and all analyses are only carried out during the second year of the simulation.





**Table 2.** Model configuration for the WRF tropical channel simulations

| *Physical setting* | *Remark* |
|---|---|
| Horizontal resolution | 27 km |
| Vertical level | 32 |
| Boundary layer scheme | Yonsei University PBL |
| Microphysics scheme | The Kessler scheme cloud microphysics scheme |
| Cumulus parameterization scheme | A modified version of the Kain and Fritsch (1990) |
| Radiative scheme | Rapid Radiative Transfer Model (RRTM) scheme for both longwave and shortwave radiation (Mlawer et al., 1997) |
| Time step | 30 seconds |
| Lateral boundary conditions | The zonal boundary condition is set to be periodic whereas be open in the meridional direction |
| Ocean coupling | 1D slab model |
| Ocean relaxation time | $1\,m$ |
| Ocean mixed layer depth | $\tau = 30\,days$ |

Starting from the second year of the model integration, we implemented a specific design to constantly excite the tropical region by adding either EK or ER wave perturbations to the model state at a regular interval during the course of the model

simulation. These external wave perturbation forcings will help examine if any PSW can develop and how these PSWs can help organize TC clusters as presented in Section 2. Note that such an external stirring mechanism is needed herein to imitate large-scale forcings in our aqua-planet simulations, similar to the terrain-induced forcing in the presence of realistic topography. For the sake of implementation in this study, these external ER and EK perturbations were added to the WRF model state at a prescribed interval of every 10 days during the second year of the model integration. Our sensitivity simulations for a range

of external wave forcing intervals showed no significant impacts (see V20). As such, we fixed a frequency of 10 days for the external wave forcing in both ER and EK experiments herein.

To take into account the first-order effects of SST climatology on the tropical atmosphere, we initialized the WRF simulations with an SST profile for which SST peaks at the equator and gradually decreases with latitudes following the same profile as in V20. Because of the ocean coupling, note that this SST distribution varies seasonally during the course of the model simulation

with the peak SST oscillating between 10S-10N (see Figure 4 in V20). This seasonally-varying SST distribution allows the tropical atmosphere to capture the migration of the ITCZ between the summer and winter seasons as expected. In the absence of landmass, the ITCZ is considered to be the key in producing conducive environmental conditions for global TCG, and so it is important to maintain its seasonal migration.

Given the output from these tropical channel simulations, several standard criteria for tracking model vortex centers are

employed, which include the minimum central pressure, the maximum vorticity and surface wind speed, warm core, and TC





**Table 3.** Threshold criteria for vortex tracking detection

| Model resolution | Latitude range | Gale-force wind ($\mathrm{ms}^{-1}$) | Minimum pressure (hPa) | Storm outer size (km) | Distance between Pmin and vorticity centers (deg) | Temperature anomaly at 400 hPa (K) | Duration (days) |
|---|---|---|---|---|---|---|---|
| 27 km | -45 S to 45 N | 17.5 | 1004 | 300 | 5 | 3 | 3 |

lifetime (see Table 3 for all vortex tracking criteria used in this study). Note that the maximum surface wind threshold is set to be 12 m s$^{-1}$ instead of 17 m s$^{-1}$ due to the 27-km resolution model output. So long as a TC vortex is detected, its first location is always defined and used for our TCG distribution.

Because our focus here is on the *relative difference* between the EK and ER waves in global TCG, subtle issues in TC
tracking related to model resolution, physical schemes, or tracking criteria sensitivity as discussed in, e.g., Zhao et al. (2009); Camargo and Zebiak (2002); Horn et al. (2014) are not considered. Similar to V20, the key characteristics of global TCG are sufficiently robust among those sensitivity experiments, so long as the comparison is relative between these two types of tropical waves and the same tracking algorithm/model configurations are used for both experiments. From this perspective, the relative differences between ER and EK experiments could offer new insights into the roles of large-scale tropical waves that
we wish to present in this study.

### 3.4 Numerical Results

To have a broad picture of how EK and ER perturbations affect the large-scale tropical environment, Fig. 2 shows a snapshot of the low-level convergence from the ER and EK experiments for a range of zonal wavenumber $m = 5 - 15$ averaged during the July (day $180 - 210$) of the second year of the simulations. One notices in Fig. 2 several key differences between these two
wave experiments. First, the ER experiment could display well a wave-like pattern at the planetary scale throughout July, so long as the zonal wavenumbers $m < 15$. In contrast, the EK experiments could not capture any wave-like structure, even when we force the model with the EK waves more frequently for the whole range of $m$.

Second, the wave-like structures in the ER experiments are also zonally stationary with time during the course of the simulations, not just for any particular moment. To further see this zonal stationarity of these wave-like patterns in the ER experiment,
Fig. 3 displays the snapshots of low-level convergence at a 5-day interval during July for one example wavenumber $m = 5$ in the ER experiment. It is seen indeed that these large-scale patterns are well maintained at all times, which is valid also for other





**Figure 2.** Horizontal cross-sections of 30-day averaged convergence at 900 hPa (shaded, unit $10^{-5}s^{-1}$) for (a)-(f) ER waves, and (g)-(l) EK waves obtained from $t = 180 - 210$ day during the second year of the WRF simulation with $m = 5, 7, 9, 11, 13, 15$.







**Figure 3.** Time evolution at a 5-day interval from day 180-210 during the second year of simulation of the low-level convergence at 900 hPa (shaded, unit $10^{-5}s^{-1}$) for (a)-(g) ER wave experiment with $m = 5$, and (h)-(n) EK wave experiment with $m = 5$.





zonal wavenumbers. For the EK experiment, the same analyses show however that the EK waves fade away quickly after being introduced to the model, leaving behind very little signal of wave-like structures as shown in Fig. 3.

Our examination of different months, variables, and vertical levels captures the same characteristics of the wave-like struc-
tures as well as their stationarity, which are all apparent only in the ER experiments. While these wave-like patterns are zonally stationary as expected for PSWs, we note that they are seasonally shifted between the northern and southern hemispheres due to the migration of the ITCZ (Fig. 4). This highlights the role of planetary-scale tropical dynamics in setting up favorable zones for PWSs to exist as obtained from our ITCZ model in Section 2.

The fact that ER waves with their maximum off the equator could develop a stationary structure during the entire course of
the simulation while EK waves could not is noteworthy. First, this result confirms that not every type of wave perturbation can develop and maintain its stationary structure, consistent with the theoretical results presented in Section 2. Specifically, tropical waves with their peaked amplitudes right at the equator would not permit a stationary structure with time. In this regard, both the WRF simulations and the ITCZ model suggest that the tropical region could possess some intrinsic dynamics that support PSWs if properly forced.

Second, the planetary-scale stationary structures obtained in the ER experiment have a profound effect on the environmental conditions that control the formation of TCG clusters. To see this, Fig. 5 shows the distribution of TCG and the corresponding TC tracks in the ER and EK simulations for different $m$. It is of significance to notice that the ER experiments could produce very specific hot spots for TCG at all wavenumbers, even in the absence of landmass surfaces or zonal SST variations. So long as the WRF model could establish the wave-like stationary structure, TCG clusters will emerge within the low-pressure
areas of the stationary structure, which can be maintained throughout the TC season in each hemisphere (Figs. 6-7). For higher zonal wavenumbers ($m > 15$), TCG clusters tend to be closer together and so the clustering characteristic gradually disappears, even in the ER experiments. This behavior again accords with our theoretical results on the upper bound of possible PSWs as presented in the previous section, and reveals the potential role of PSWs in governing TCG clusters beyond the land-surface effects.

Consistent with the faint wave-like structure in the EK experiments, Fig. 5 displays a more homogeneous TCG distribution in all EK experiments. Within the tropical domain, TCs essentially form anywhere, regardless of the EK zonal wavenumbers. Similar to the results in V20, we observe that EK waves quickly dissipate upon being introduced into the model and leave behind no clear footprint on the clustering of TC formation. This result also accords with the previous aqua-planet simulations in which TCs can basically form anywhere, without any apparent TC clusters (e.g., Merlis et al., 2013; Rauscher et al., 2013;
Merlis et al., 2013; Vu et al., 2021).







**Figure 4.** Similar to Figure 3 but for January.





**Figure 5.** Distribution of the TCG location during the second year of the WRF simulations (blue/red dots are for the December-April/May-November season) for (a)-(f) ER wave experiments; and (g)-(l) EK wave experiments with $m = 5, 7, 9, 11, 13, 15$. Black solid lines denote the TC tracks corresponding to each TCG.



Because of the seasonal migration of SST, we note that TC clusters are not fixed but move along the low-pressure areas associated with the planetary-scale wave patterns. As seen in Figure 5, such seasonality of TC clusters and PSWs are well applied for all zonal wavenumbers in the ER experiments, except for $m = 5$ that shows a somewhat mixed TC development in both hemispheres during the boreal winter/summer seasons. While we dont have satisfactory explanation for $m = 5$, the persistent alignment of TCG clusters and PSWs in the ER experiments for all other zonal wavenumbers clearly demonstrates how environments associated with these planetary-scale waves can help create favorable "hot spots" for global TCG in the aqua-planet setting.

In the real atmosphere, it should be noted that terrain forcing often acts as a driver for planetary-scale structures (e.g., Charney and DeVore, 1979; Legras and Ghil, 1985; Kieu et al., 2023). These terrain-induced mechanisms may impose specific modes on tropical dynamics and generate similar hot spots for global TCG beyond the direct land-surface or SST zonal anomaly effects. While the role of such planetary-scale terrain forcing in the clustering of global TCG is currently not well understood, the role of PSWs as captured in our herein WRF simulations could at least reveal the importance of such internal dynamics of the large-scale circulation, which seems to be consistent with the previous analyses of tropical waves and TCG. In fact, a study by Molinari et al. (2007) about the effect of ER wave packet on TCG found that a wave packet of wavelength 3600km tends to induce TCG within one-half wavelength to the east of a bandpass-filtered ER wave as well as an extended region of bandpass-filtered cyclonic vorticity and active convection. Molinari et al. (2007) further proposed a conceptual model that provides a sequence of steps for TCG associated with ER wave packet, which emphasizes the role of long ER waves in supporting the development of convection within the stationary convergent background In this regard, our modeling and analytical results herein support the potential role of ER waves in global TCG climatology.

## 4 Conclusion

In this study, the clustering of global tropical cyclogenesis (TCG) was examined from both theoretical and numerical perspectives, using a simplified ITCZ model and idealized tropical channel simulations. By considering TCG clusters as a result of the tropical environment, our main aim of this work is to search for the possible existence of stationary planetary-scale structures in the tropics that can control global TCG clusters. Building on the previous analyses in Wang et al. (2019), our theoretical investigation of the ITCZ breakdown model captured a number of important results related to the existence of tropical wave-like stationary modes.

First, we obtained an upper bound on the zonal wavenumbers for the planetary-scale stationary structures. For a typical Earth atmosphere, this upper bound on the zonal wavenumber is $m < 11$ beyond which no stable stationary modes can exist. The




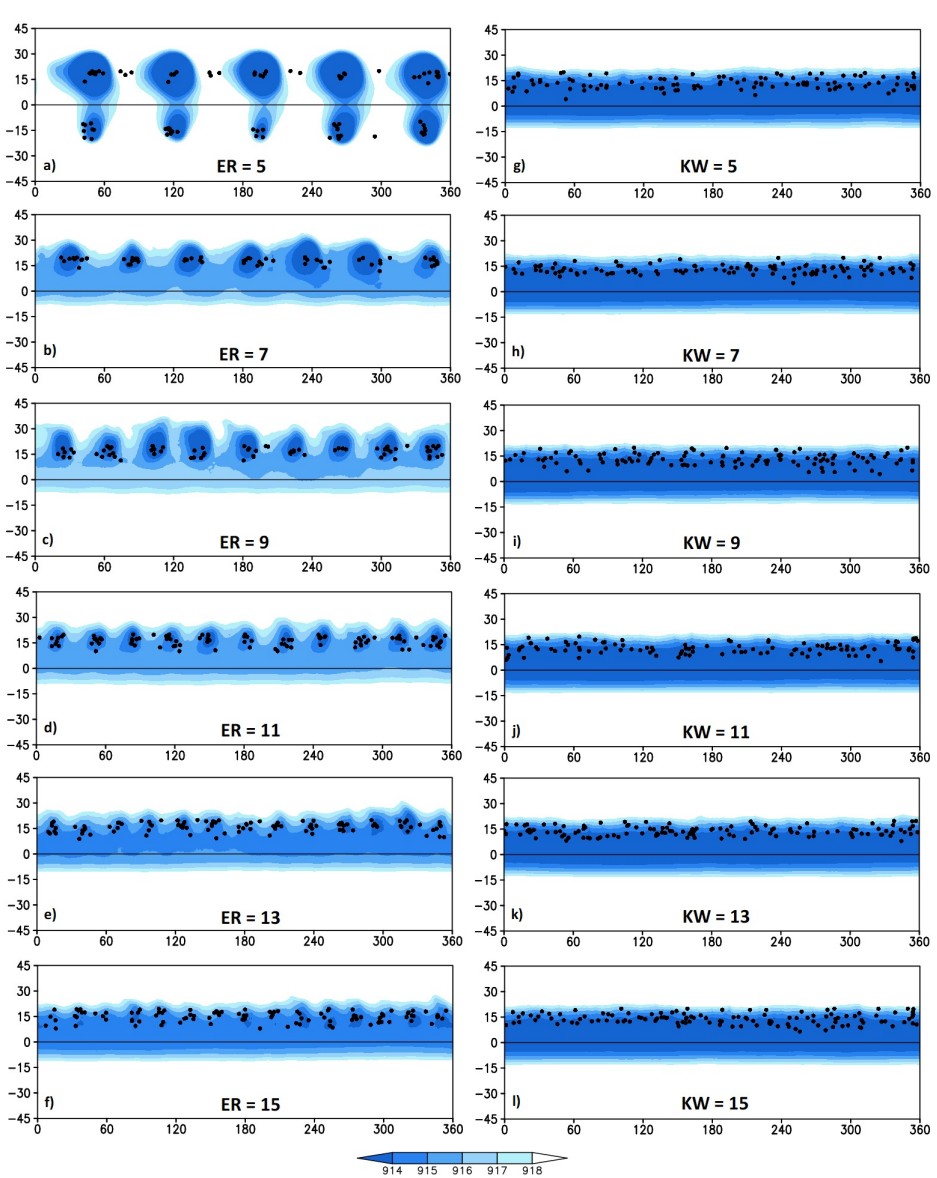

**Figure 6.** (a)-(f) Pressure at 850 hPa (shading) embedded with initial TC genesis locations (black dots) during the summer in the Northern Hemisphere from July-October for (a)-(f) ER wave experiments; and (g)-(l) EK wave experiments with $m = 5, 7, 9, 11, 13, 15$.



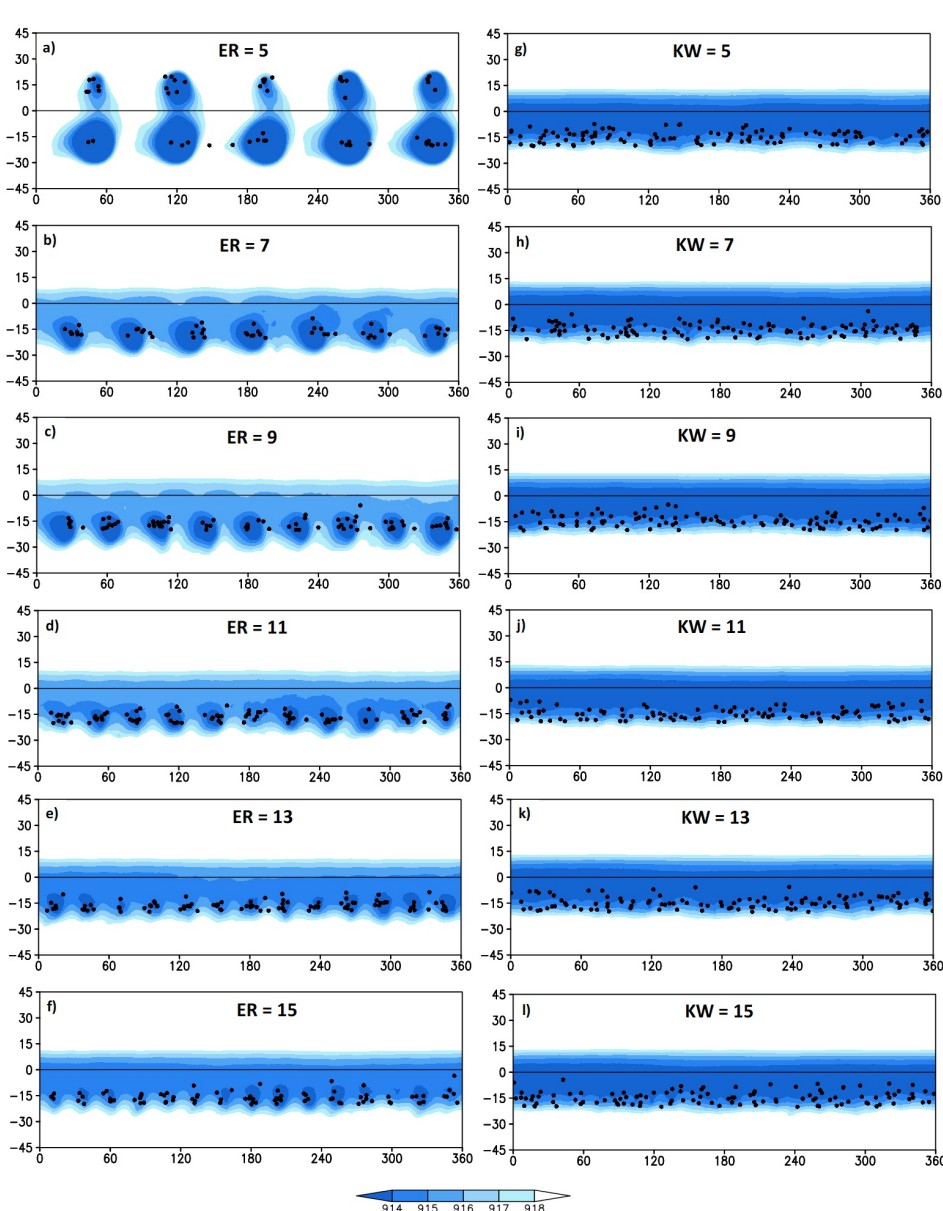

**Figure 7.** Similar to Figure 6 but during the summer in the Southern Hemisphere from January-April





exact zonal wavenumbers that can develop such a stationary structure depend further on other factors such as the magnitude of
zonal mean flows, the Rossby number, the Ekman number, or the Rayleigh number, which vary with season and global climate.
However, the existence of such an upper bound limit is itself of importance, as it indicates what planetary-scale structure can
possess stationary modes.

Second, our analytical results also suggest some specific meridional structures that could support stationary wave modes
in the tropics. Under the Kolmogorov forcing, these stationary structures must have a maximum amplitude away from the
equator, thus eliminating a number of tropical waves in developing a stationary state. These important results obtained from
the ITCZ model were indeed validated by our full-physics simulations with the WRF model. Using two common tropical waves
including the equatorial Rossby (ER) and equatorial Kelvin (EK) waves in a range of WRF simulations, it was found that ER
waves could generate a stationary structure for zonal wavenumbers $m \in [5..13]$. For larger wavenumbers, these stationary
structures gradually fade away, consistent with the upper bound result obtained from the ITCZ model. In contrast, EK waves
with their wave amplitude peaked at the equator could not hold their stationary structures, regardless of zonal wavenumbers or
how frequently these waves are added to the model state. These outcomes from the WRF simulations confirm our theoretical
results and highlight the subtle nature of tropical dynamics in producing planetary-scale stationary waves.

A particular consequence of the planetary-scale stationary modes is the formation of global TCG clusters in the tropical
region, even within the aquaplanet setting. For the ER wave experiment, a group of TCG clusters could emerge in the absence
of all land surfaces or SST zonal anomalies. Within the low-pressure areas associated with these stationary structures, the
favorable conditions for TCG such as strong low-level moisture convergence, enhanced vertical motion, or stronger divergence
at the upper level were all observed, which provide an ideal environment for TCG. As a result, TCs tend to form distinctly within
the large-scale low-pressure areas associated with stationary waves. With the seasonal migration of the ITCZ, we observed also
that these TC clusters shift seasonally between the northern and southern hemispheres as expected.

The implication of global TC clusters obtained from these aqua-planet WRF simulations is noteworthy. From the dynamical
standpoint, our finding suggests that the "hot spots" for global TC formation can be a manifestation of tropical internal dy-
namics, which contains planetary-scale waves that set up different favorable conditions for TCG, even in the absence of zonal
SST variations. Note that these planetary-scale waves, which may be forced by large-scale forcings such as terrain, differ from
the typical tropical easterlies whose horizontal scale is of $\sim \mathcal{O}(10^5)$ km (Dunkerton and Baldwin, 1995; Sobel et al., 2004;
Holton, 2004). From this regard, the important role of the tropical atmosphere in the clustering of TC formation must be taken
into account beyond the direct alignment of zonal SST anomalies as also emphasized recently in Kieu et al. (2023).



Given the nature of our approaches and analyses in this study, a number of drawbacks should be mentioned. First, our theoretical analyses based on the ITCZ model could not specifically indicate which zonal wavenumbers will be stationary. All we could obtain herein is that there is an upper bound on the possible stationary zonal wavenumber and structure, similar to

the dynamical transition discussed in Wang et al. (2019). While these results are still of sufficient interest from a theoretical standpoint, quantifying the exact number of TC clusters at the global scale is also of significance but would require much more in-depth analyses of tropical dynamics beyond the stationary wave structure presented herein. In particular, the two-dimensional forcings used in our ITCZ breakdown model could not fully represent the three-dimensional dynamics of the ITCZ. As such, some inherent weaknesses of our ITCZ breakdown model due to these simplifications are anticipated.

Second, our WRF simulations were limited to a tropical channel simulation with a simple ocean coupling. While our sensitivity experiments with different resolutions and physical parameterization schemes captured the same large-scale characteristics as presented in Vu et al. (2021), it is not known if more complex ocean coupling or other modeling systems could replicate any of these above results. Nonetheless, the fact that the WRF model in this study could develop such a coherent stationary structure in all of our experiments could at least indicate that the tropical dynamics do contain some intrinsic modes that dictate

the large-scale formation of TCs in the absence of all landmass surfaces. More cross-validations with other modeling systems are needed to ensure that these stationary modes are physical and applicable to global TC formation.

**Appendix A:  Upper bound on zonal modes**

Following W19, we substitute (12) into Eq. (5) to obtain the conditions for the amplitude of stationary waves as follows

$$\sum_{n \geq 0} B_{m,n} A_{m,n} (E\pi^2 A_{m,n} + I) |\phi_{m,n}|^2 = 0, \tag{A1}$$

$$\sum_{n \geq 1} D_{m,n} \widetilde{A}_{m,n} (E\pi^2 \widetilde{A}_{m,n} + I) |\widetilde{\phi}_{m,n}|^2 = 0, \tag{A2}$$



where the coefficients $A_{m,n}, B_{m,n}, C_{m,n}, D_{m,n}, E_{m,n}$ are

$$
\begin{cases}
A_{m,n} = a^2 m^2 + (n+1/2)^2 \\
B_{m,n+1} = (1 - A_{m,n+1}) \qquad\qquad , \quad n \geq 0, \quad |m| \geq 1, \\
C_{m,n} = \frac{2\pi^3 E A_{m,n}^2 + 2\pi I A_{m,n} - 2iam}{am\pi^2 R}
\end{cases}
$$

$$
\begin{cases}
A_{0,n} = (n+1/2)^2 \\
B_{0,n+1} = (1 - A_{0,n+1}) \qquad , \quad n \geq 0, \quad m = 0, \\
C_{0,n} = 2\pi^3 E A_{0,n}^2 + 2\pi I A_{0,n}
\end{cases}
$$

(A3)

$$
\begin{cases}
\widetilde{A}_{m,n} = a^2 m^2 + n^2 \\
D_{m,n} = (1 - \widetilde{A}_{m,n}), \qquad\qquad , \quad n \geq 1, \quad |m| \geq 1, \\
E_{m,n} = \frac{E\pi^3 \widetilde{A}_{m,n}^2 + \pi I \widetilde{A}_{m,n} - i2am}{am\pi^2 R}
\end{cases}
$$

$$
\begin{cases}
\widetilde{A}_{0,n} = n^2 \\
D_{0,n} = (1 - \widetilde{A}_{0,n}), \qquad\qquad , \quad n \geq 1, \quad m = 0. \\
E_{0,n} = E\pi^3 \widetilde{A}_{0,n}^2 + \pi I \widetilde{A}_{0,n}
\end{cases}
$$

(A4)

By imposing the physical requirement on the existence of the wave-like solution, i.e., $\phi_{m,n} \neq 0$ and $\widetilde{\phi}_{m,n} \neq 0$, the conditions (A1)-(A2) can provide a great insight into the stability and structure of the stationary wave solutions.

Indeed, one notes first from (A3) that the coefficients $A_{0,n} > 0$, $B_{0,n} \leq 0$ and $D_{0,n} \leq 0$. Thus, it can be directly seen from the quadratic form of (A1) that for $m = 0$

$$
\phi_{0,n} = 0, \ \ \widetilde{\phi}_{0,n} = 0, \qquad n \geq 0,
$$

and so there would exist no stationary waves for zonal wavenumber $m = 0$.

For $m \neq 0$, we note from (A1) that all possible stationary waves with $m \neq 0$ must satisfy the following conditions

$$
\begin{cases}
\phi_{m,n} = 0, \text{ when } \frac{\sqrt{3}}{2k} \leq a < \frac{\sqrt{3}}{2k-2}, \ |m| \geq k, \ k = 1, 2, 3... \\
\widetilde{\phi}_{m,n} = 0, \ n \geq 1, \text{ for all } \quad a > 0.
\end{cases}
$$

(A5)





This can be seen by recalling that $A_{m,n} = a^2 m^2 + (n+1/2)^2 > 0 \forall m, n$, and $B_{m,n} = (1 - a^2 m^2 - (n+1/2)^2) < 0$ for $\forall n \geq 1$. The only possibility for the amplitude $\phi_{m,n} \neq 0$ is when at least one of $B_{m,n}$ is positive, which can only be true for $n = 0$ and a sufficiently small values of $m$. Indeed, by applying the condition $B_{m,n} < 0$ with $n = 0$, it is easy to see from Eq.(A1) that $\phi_{m,n} = 0$ for all sufficiently large values of $m > k$, regardless the value of $n$. This lower bound $k$ for the wavenumber $m$ can be found by looking at the maximum value of $B_{m,n}$ at $n = 0$ and imposing a condition that $B_{k,0} \leq 0$ but $B_{k-1,0} > 0$ for this lower bound wavenumber $k$. That is,

$$B_{k,0} = \frac{3}{4} - k^2 a^2 \leq 0 \text{ and } B_{k-1,0} = \frac{3}{4} - (k-1)^2 a^2 \geq 0 \tag{A6}$$

from which one obtains $\frac{\sqrt{3}}{2k-2} \geq a \geq \frac{\sqrt{3}}{2k}$, similar to what found in W19 for the dynamical transition of the ITCZ model. Using the same argument, it can be seen then that $\widetilde{\phi}_{m,n} = 0$ for all $(m, n)$. Given the aspect ratio $a$ in the Earth's tropical atmosphere, one can therefore find an upper bound on the possible stationary waves in the tropical channel.

The lower bound wavenumber $k$ for the zonal wavenumber has a profound implication for global TC formation. Specifically, for all zonal wavenumber $m > k$, $\phi_{m,n} = 0$ and so there exists no stationary wave mode for $m > k$. For the typical Earth's atmosphere with $a \sim 0.054$, one obtains $k \sim 11$. This imposes a strong constraint on the possible structure of stationary waves. To see the significance of this result, consider an atmosphere for which one can find $k = 2$. Assume further the environmental condition ensures a critical Rayleigh number that allows for a stationary mode $m = 2$, Eq. (A1) for the most simple case of $n = 2$ then gives

$$A_{5,0}(\frac{1}{4} - 4a^2)(E\pi^2 A_{5,0} + I)|\phi_{5,0}|^2 + A_{5,1}(-\frac{5}{4} - 4a^2)(E\pi^2 A_{5,1} + I)|\phi_{5,1}|^2 = 0 \tag{A7}$$

Since the bound $k = 2$, we must then have $\frac{1}{4} - 4a^2 > 0$ but $-\frac{5}{4} - 4a^2 < 0$, i.e., $a < \frac{1}{4}$. For this case, one can therefore obtain

$$\phi_{5,1} = \left( \frac{(1 + 16a^2)(1 - 16a^2)(E\pi^2(1 + 16a^2) + I)}{(9 + 16a^2)(5 + 16a^2)(E\pi^2(9 + 16a^2) + I)} \right)^{1/2} \phi_{5,0} \tag{A8}$$

and the stationary wave is now given by a form

$$\psi'(x,y) = \left( \phi_1 \cos(\frac{\pi y}{2}) + \phi_2 \cos(\frac{3\pi y}{2}) \right) \sin(2a\pi x) \tag{A9}$$

Note that $n = 2$ is the minimum limit that one can take in order for the planetary-scale stationary waves to exist. The wave structure for $n > 2$ can be also obtained similarly, but the key point from the above solution is that any stationary wave mode



will have to peak away from the equator due to the combination of cosine functions as discussed in the main text. As a result, global TC clusters must be located off the equator as observed.

*Data availability.* The simulated data from the WRF model can be reproduced by using the WRF model setting and initial/boundary conditions presented in Section 3. Jordan's sounding profile is taken directly from Jordan (1958).

*Acknowledgements.* This research was partially supported by the ONR Young Investigator Program and NSF/AGS program.

*Author contributions.* CK perceived the ideas and carried out theoretical analyses/experiment designs. AV conducted numerical experiments and diagnostics. Both authors contributed to the writing of this work.

*Competing interests.* The authors declare that no competing interests are present



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
