# Peer review of "Roles of Tropical Waves in the Formation of Global Tropical Cyclone Clusters"

_EGUsphere, 2023_

## Author Comment (AC1)

**Responses to Reviewer 1**

We appreciate Reviewer 1 for your thoughtful comments and detailed corrections. In this revision, we have taken into account your comments and made several changes including 1) added more discussion to distinguish this work from Wang et al. (W19) and shortened some derivations to remove repetitions, 2) added more discussion to better connect the theoretical and numerical sections, 3) provided additional information on the vertical normal mode initialization as well as a new figure (Fig. 4) to better interpret the wave stationarity, and 4) corrected different typos and inaccurate expressions that you have help pointing out. Please find below our point-to-point responses to your questions and comments (all of these changes are highlighted in the red font so you can quickly follow our revisions).

1. Section 2 is mostly repeating the equations in W19 with very little information about how this part is different from W19. If they are exactly the same, then there is no need to repeat it again as if it is a new result. If not, please clarify the differences and emphasize what's new.

Thank you for your comment on this. While this study shares the governing equation and the bulk of derivations with our previous study (Wang et al. 2019, or W19), we wish to mention that this work focuses on very different aspects of tropical dynamics and waves in global TC formation as compared to W19. Specifically, unlike W19 that examined the dynamical transition and the stability of the ITCZ model, this study focuses on the existence and the structure of planetary-scale stationary waves that can help us understand global TC clustering. In particular, the main results of this study are numerical simulations and analyses that bring new insights into the upper bound as well as the structures of stationary modes governing the global formation of TC clusters, which were absent in W19. This is the reason why we only briefly present the governing equation along with few most important steps in the main text, while moving all detailed derivations to Appendix A to avoid duplication. In this revision, we have modified our theoretical part further to better highlight the difference between W19 and our present study, and streamlined some derivations by referring directly to W19 as you suggested.

2. The connection between theoretical and modeling part is not stated clearly. The current layout gives me a very abrupt transition from Section 2 and Section 3. For example, what is the connection between the imposed ER wavenumber in WRF simulations and theoretically derived K bound; what are the associated magnitude of mean flows, Rossby number, Ekman number and Rayleigh number in the WRF simulations and the corresponding theoretical value of PSW wavenumber from the theory?

Your comment is well-taken. Although one can obtain all theoretical numbers such as the Ekman number, the Rossby number, or the Rayleigh number from model simulations, we wish to note that these theoretical numbers are derived from a highly simplified ITCZ model under idealized settings. Also, they are defined only under the strict conditions of 2D dynamics with the specific Kolmogorov forcing given by Eq. (2). This is the reason why we have to use the WRF model to further examine the stationary waves and their role in global TC clustering instead of using theoretical results. For the full-physics WRF model, the tropical dynamics is no longer reduced and so the exact value of those theoretical numbers will be less significant. What more significant is, however, the broad implication obtained from the ITCZ model, which suggests 1) an upper bound for stationary zonal wavenumbers, and 2) the meridional structure that stationary modes must possess. These properties are global dynamics and so expected to hold true from a large-scale perspective, much like the TC eyewall structure or warm core are the intrinsic properties of TCs that any model should capture, regardless of their exact magnitude. This explains why we choose 2 different tropical waves with different zonal wave number to valid these aforementioned broader results in the WRF model, but have not provided

detailed analyses of all nondimensional numbers. This comment is indeed important and so we have added more discussions about this issue in this revision. We hope it could address your concern.

Technical corrections:
1. L121-124: expand this part to clarify the differences between the theoretical analysis in this manuscript and W19.

More discussions have been added.

2. L127: subscription missing in the streamfunction expression: 'LU_0' -> 'L_yU_0'

This typo has been fixed.

3. L123: a new character 'I' is introduced, which is exactly the same as A in W19. Why not adopt the same character?

We use '$I$' here to avoid confusion with W19's expression '$A$' in the derivations presented in Appendix A.

4. L150: 'max' on the exponent is a little misleading, I though it was taking some maximum value…

We have changed 'imax \pi' to 'ima \pi x' to avoid confusion.

5. L168-169, L363: a~0.06 and K~12 in W19, please be consistent.

Thank you for your detail checks. We have changed to a~0.06 and K <= 12

6. L250 and below: V20 is not defined. And I only found Vu et al. 2021.

This is our typo. We have now corrected this citation to Vu et al. 2021 (V21) in line 251.

7. In the table 2, the last two rows are mismatched.

This has been fixed.

8. L271-272: I think the purpose of the paper is to show with no land it is possbile to have PSW by earth internal dynamics, but here it reads very contradictory.

Our discussion here was unclear. What we really meant is that the external wave forcing is needed to excite tropical waves in the aqua-planet setting. Without realistic terrain, this external wave forcing is a reasonable mechanism that we can force the model to see whether these waves can maintain their stationary structure or not. This paragraph has been revised to make it clear.

9. L275: 'no significant impacts' on what?

We have added 'no significant impacts on TC formation' to this sentence.

10. The criteria in Table 3 are used according to Vu et al. 2021, should state this in text 'refer to Vu et al. 2021 for more details'

This clarification has been added.

11. For the numerical results, are the EK results the same as the ones in Vu et al. 2021?

The EK simulations in this study have been designed to match with Vu et al. 2021, and so yes we obtain similar results for EK waves as in V21.

12 Do the ER/EK waves imposed in the WRF model have seasonality? Can you show the seasonality?

Thank you for the question. We do see the seasonality of both ER/EK waves stirring mechanisms as shown in Figs 5, 6 and 7 in the manuscript. Please note that red dots in Fig. 5 correspond to TCs that form in the Northern Hemisphere summer, while blue dots are for the Southern Hemisphere summer. We have added more discussions here to better highlight the TC seasonality from our aqua-planet configuration.

13. Figure 5: it's very interesting to see the TC numbers differ in different ER experiments while the number keeps quite stable in EK experiments. Can you discuss a little bit why?

For EK wave experiments, the wave structures quickly dissipate upon being introduced into the model, which can be seen in Fig. 2. Therefore, TCs can form anywhere within the tropical region, and different wavenumbers of EK waves do not seem to influence the total global number of TCs.

For ER wave experiments, the wave structures do not fade way but they are well-maintained after introduced into the model. These different waves generate different hot spots for TCs to form (cf. Figs. 6 and 7). As such, the locations for TC formation are now governed by ER waves, leading to different total global number of TCs for different zonal wavenumbers. We have added this discussion in the revised version per your comment.

14. L339: missing ' in 'don't have'

This typo has been fixed. Thank you again for your comments.

---

## Author Comment (AC2)

**Responses to Reviewer 2**

We wish to thank Dr. Paul Roundy for your constructive comments. In this revision, we have followed your suggestions and modified our work as follows: 1) provided additional information on the vertical normal mode initialization and added new Figure 4 to better interpret the wave stationarity as you suggested, 2) added more discussion to distinguish this work from our previous work (Wang et al. 2019) and streamlined some of our derivations to avoid redundancy, 3) included further discussion to connect our theoretical and numerical sections, and 4) corrected several typos and/or inaccurate expressions. Below please find our point-to-point responses to your concerns. For your convenience, all of our modifications are highlighted in the red font so you can quickly follow our changes in this revision.

1. Although vertical normal modes are a popular way to build simple models of convectively coupled waves, in the real atmosphere and in numerical models, they may not exist independent of the effects of coupling of waves to convection (which drives overturning circulations limited in vertical extent at the tropopause). Real waves, even convectively coupled ones, typically propagate vertically and the tropopause is not a limit to their movement. This fact implies that initializing a model with idealized normal mode waves will result in the model having to move toward a state consistent with its internal dynamics. This point might not refute the authors' overall arguments because initializing the model with a wave disturbance of the same type but more consistent with the model's native form of the wave, might still result in a similar outcome to what they showed.

Thank you for your insightful about the vertical normal mode for tropical waves. Yes, we are aware of this issue and in fact have adopted the same approach as in our early study (Vu et al. 2021) in which we have tried different wave initializations and run the model for 1 year as a model spinup so that the model could establish its own dynamics consistent with the physical options, boundary conditions, and domain setting. All of the analyses are then carried from the second year such that the impacts of normal mode initialization are minimized. As reported in Vu et al. (2021), this 1-year spinup is sufficiently robust in the sense that the key annual properties of the tropical atmosphere are consistent among subsequent years, regardless of period we have analyzed. In this regard, the issue with vertical normal mode adjustment is expected to be transitional and may have small effects on the overall model outputs as you also noted.

Another way to address this issue more conclusively is to use a very high model top along with many different vertical resolutions to examine the model sensitivity. This approach requires however additional sensitivity analyses such as different vertical wave profiles or model top boundary conditions to ensure the model stability, which are however beyond our computational resources at present. In this revision, we have provided in Section 3 some additional discussions related to vertical normal modes so readers are aware of this potential problem in our numerical simulations.

2. The authors' analytical solution suggests that stationary waves might occur in the tropics, if the assumptions of the simple model apply in nature. Yet nature can yield stationary waves through other mechanisms. The leading one is probably forcing by regional SST anomalies, interaction with topography, etc., which their model set-up would not include (as the authors already explain). Other possible sources of stationary waves include waves that would otherwise propagate, but whose propagation is balanced by advection. In a model environment that includes a steady background state flow, it's conceivable that such signals could occur with stability. In nature, this kind of steady basic state is implausible, because sea surface temperature patterns vary over time. I recommend that the authors analyze their model basic state for conditions that could lead to such stationary advection-balanced propagation for Rossby waves. It may be that the model background flow explains why Rossby waves can become stationary in the model but Kelvin waves cannot. In order for Rossby waves

to be stationary under conditions of balance by advection, they must be non-dispersive. This would place a control on which scales of the waves would be favored by this mechanism to become stationary.

We agree. In fact, there are several different pathways to trigger large-scale tropical waves such as terrain forcings, land-sea interaction, or ocean coupling that we are not able to capture with the theoretical and numerical models presented in this study. We did mention these potential mechanisms in our previous version (see, e.g., the second to the last paragraph of Section 1 or the last paragraph of Section 3 in the original submission), which explains why we have to introduce the Kolmogorov forcing to mimic those external pathways that the aqua-planet settings cannot capture.

One thing that we wish to take this opportunity to clarify further is that while the ITCZ model could suggest a potential stationary mode as shown in our study, there is no guarantee that this mode must exist in nature due to different large-scale factors that could break the stationarity as you correctly pointed out, even within the idealized framework. In fact, this study is built on our previous works (Wang et al. 2019, Pan et al. 2021), which presented more detailed analyses of the dynamical transition, stability, and bifurcation of stationary solutions. Any change in background flows, ocean coupling, or tropical wave types could break the stationarity, which is why we use the WRF model to search for the stationary modes as presented in this study, instead of relying entirely on the theoretical results. To our knowledge, the modelling approach is the best way to look for such stationary modes in the presence of full physics.

Regarding your suggestion of analyzing background flow for the EK and ER experiments, we concur that the background flow is one of the key factors that decide the stability of stationary solutions (this can be seen directly from the Rayleigh number defined below our Eq. 5). Any change in the background will modify this Rayleigh number and lead to the transition to a new state. Per your comment, we have now added Fig. 4 in this revision, which compares the mean flows between ER and EK experiments. As can be seen in the new Fig. 4, EK waves are indeed located in purely positive gradient zonal wind regime (i.e., $\partial \overline{U}(y,t)/\partial y > 0 \ \forall y$) , which does not satisfy Rayleigh's barotropic instability criterion. Thus, all EK waves introduced into the model cannot intensify. ER waves, on the other hand, exist in the barotropically unstable zone ($i.e., \exists \ y_0 \mid \partial^2 \overline{U}(y_0,t)/\partial y^2 = 0$), and so it supports the wave growth according to Rayleigh's theorem. That is, any disturbance introduced into this background could grow and maintain its structure subsequently.

Of course, the instability of any stationary mode depends not only on the mean state but also on other parameters such as latitude, level, or convergence/divergence (in our ITCZ model, the convergence/divergence is represented by the Kolmogorov forcing). Because the stability and related dynamical transition of stationary modes have been examined in much more details in our previous works, we don't repeat them here. The key point that we wish to highlight in this study is that the wave instability is realized only for certain waves with low zonal wavenumbers, which are captured in our ER experiments and support the clustering of global TC formation. This discussion has been now included in our revision per your suggestion. We hope that this could address your concern and thank you again for your comments and insights.